# Critical role of *Babesia bovis* spherical body protein 3 in ridge formation on infected red blood cells

Atefeh Fathi[1,2], Hassan Hakimi[3ʘ], Miako Sakaguchi[4ʘ], Junya Yamagishi[5], Shin-ichiro Kawazu[1], Masahito Asada[1] *

1 National Research Center for Protozoan Diseases, Obihiro University of Agriculture and Veterinary Medicine, Obihiro, Hokkaido, Japan, 2 Graduate School of Animal and Veterinary Science and Agriculture, Obihiro University of Agriculture and Veterinary Medicine, Obihiro, Hokkaido, Japan, 3 Department of Veterinary Pathobiology, College of Veterinary Medicine and Biomedical Sciences, Texas A&M University, College Station, Texas, United States of America, 4 Central Laboratory, Institute of Tropical Medicine (NEKKEN), Nagasaki University, Nagasaki, Nagasaki, Japan, 5 Division of Collaboration and Education, International Institute for Zoonosis Control, Hokkaido University, Sapporo, Hokkaido, Japan

ʘ These authors contributed equally to this work.
* masada@obihiro.ac.jp

**Data Availability Statement:** All relevant data are within the manuscript and its Supporting Information files.

## Abstract

*Babesia bovis*, an apicomplexan intraerythrocytic protozoan parasite, causes serious economic loss to cattle industries around the world. Infection with this parasite leads to accumulation of infected red blood cells (iRBCs) in the brain microvasculature that results in severe clinical complications known as cerebral babesiosis. Throughout its growth within iRBCs, the parasite exports various proteins to the iRBCs that lead to the formation of protrusions known as "ridges" on the surface of iRBCs, which serve as sites for cytoadhesion to endothelial cells. Spherical body proteins (SBPs; proteins secreted from spherical bodies, which are organelles specific to Piroplasmida) are exported into iRBCs, and four proteins (SBP1-4) have been reported to date. In this study, we elucidated the function of SBP3 using an inducible gene knockdown (KD) system. Localization of SBP3 was assessed by immunofluorescence assay, and only partial colocalization was detected between SBP3 and SBP4 inside the iRBCs. In contrast, colocalization was observed with VESA-1, which is a major parasite ligand responsible for the cytoadhesion. Immunoelectron microscopy confirmed localization of SBP3 at the ridges. SBP3 KD was performed using the *glmS* system, and effective KD was confirmed by Western blotting, immunofluorescence assay, and RNA-seq analysis. The SBP3 KD parasites showed severe growth defect suggesting its importance for parasite survival in the iRBCs. VESA-1 on the surface of iRBCs was scarcely detected in SBP3 KD parasites, whereas SBP4 was still detected in the iRBCs. Moreover, abolition of ridges on the iRBCs and reduction of iRBCs cytoadhesion to the bovine brain endothelial cells were observed in SBP3 KD parasites. Immunoprecipitation followed by mass spectrometry analysis detected the host Band 3 multiprotein complex, suggesting an association of SBP3 with iRBC cytoskeleton proteins. Taken together, this study revealed the vital role of SBP3 in ridge formation and its significance in the pathogenesis of cerebral babesiosis.

**Funding:** This work was primarily supported by grants from Japan Society for the Promotion of Science (https://www.jsps.go.jp/english/index.html) to M.A. (21KK0121, 22K05982), H.H. (15K18783, 19K15983), and S-I.K. (19H03120, JPJSBP120212501). The funders had no role in study design, data collection and analysis, decision to publish, or preparation of the manuscript.

**Competing interests:** The authors have declared that no competing interests exist.

## Author summary

*Babesia bovis* causes a high-mortality complication called cerebral babesiosis in cattle, similar to cerebral malaria in humans. Both complications are caused by the cytoadhesion of infected red blood cells (iRBCs) to the host brain endothelial cells. These parasites export numerous proteins to the host iRBCs and produce protrusions on the iRBCs that are called ridges for *B. bovis* and knobs for *Plasmodium falciparum*. Ridges and knobs play an important role in cytoadhesion as they are the sites of adherence; however, the molecules responsible for ridge formation remain unknown. In this study, we showed that SBP3 is a crucial protein for ridge formation. The SBP3 knockdown parasites showed severe growth defects and abolition of ridges on the iRBCs, and cytoadhesion of iRBCs to the bovine brain endothelial cells was significantly reduced. An immunoprecipitation experiment suggested an association of SBP3 with the host Band 3 multiprotein complex. Although there is no similarity in amino acid sequences, we suggest SBP3 to be a functional analog of KAHRP in *P. falciparum*. In summary, our results shed light on the molecular mechanism of ridge formation and the pathogenesis of *B. bovis*.

## Introduction

*Babesia* is a tick-transmitted intraerythrocytic protozoan parasite infecting a large variety of domestic and wild animals, and sometimes humans [1]. Although more than 100 species of *Babesia* are known to infect mammalians, bovine *Babesia* parasites cause serious economic loss to cattle industries around the world [1,2]. Current control measures against bovine babesiosis are limited. Few drugs are available, and there are limitations in the usage of vaccines; moreover, resistance of the vector tick to acaricides has also become apparent recently [1–3]. Therefore, exploration of new control and treatment measures for the disease is urgently needed.

The two major species that cause bovine babesiosis are *Babesia bovis* and *B. bigemina*. The main clinical symptoms of bovine babesiosis are fever, anemia, and hemoglobinuria. While these symptoms are mainly caused by intravascular hemolysis and are common for *B. bigemina* infection, *B. bovis* infection adds an additional complication called cerebral babesiosis. Cytoadhesion of *B. bovis*-infected red blood cells (iRBCs) to brain microvascular endothelial cells leads to the sequestration of blood flow in the brain vascular system, causing cerebral symptoms and a high fatality rate in infected animals [1,4,5].

Protrusions called "ridges" on the surface of iRBC are reported as the main sites for cytoadhesion to endothelial cells. Ridges are unique structures for *B. bovis*-iRBCs, and currently it is known that multiple proteins are exported from the parasites to the cytoplasm and surface of the iRBCs [6–14]. Although the function and export mechanisms are unknown for most of these proteins, it is thought that they contribute to enhancement of metabolite exchange, increase of RBC rigidity, and cytoadhesion to the host endothelial cells [4,7,15]. It was speculated or partially shown that two parasite proteins, variant erythrocyte surface antigen 1 (VESA-1) and spherical body proteins (SBPs), which are localized near or on the surface of iRBCs, have a function in ridge formation and cytoadhesion [4,10,15].

VESA-1 are found as heterodimeric proteins encoded by the expanded multigene family known as *ves-1α* and *ves-1β* [10]. These proteins localize on the surface of ridges and form a dynamic structure that displays antigenic variation and facilitates host immune evasion [10,15]. Furthermore, VESA-1 is the only known ligand for cytoadhesion to the endothelial

cells [15–17]. Meanwhile, four SBPs (SBP1-4) are known in *B. bovis*. They are localized in spherical bodies, an organelle specific to Piroplasmida that is equivalent to dense granules in other apicomplexan parasites, exported to iRBCs cytoplasm, and finally allocated near the surface of the iRBC. It was previously reported that gene coding for SBP1, 3, and 4 are single-copy, whereas the gene coding for SBP2 is a multi-copy consisting of one complete and 12 truncated copies [8,9,11,12]. Although the function of SBPs has not yet been fully understood, their localization suggested a function associated with ridge formation [4]. Gallego-Lopez *et al.* recently reported that overexpression of the SBP2 truncated copy 11 (SBP2t11) reduced the cytoadhesion of iRBCs to bovine brain endothelial cells (BBECs) [18]. In contrast, the cytoadhesion-related functions of other SBP proteins remain unknown.

In this study, ridge formation and the cytoadhesion-related functions of SBP3 were elucidated using Myc-tagged SBP3-expressing parasites (SBP3-Myc) and SBP3 *glmS*-inducible knockdown parasites (SBP3-*glmS*). The detailed localization of SBP3 and its association with VESA-1 and SBP4 were analyzed. SBP3-*glmS* were evaluated for their ridge formation and cytoadhesion of iRBC to BBECs.

## Results

### SBP3 and VESA-1 colocalize at the ridges of iRBCs

To investigate the localization and timing of SBP3 export, SBP3-Myc, which was produced in our previous study [19], was stained with anti-Myc and anti-VESA-1 antibodies, and the parasite was also stained with anti-SBP4 antibody. The SBP3 signal was either scarcely or not detected at all in the single-form parasite (Figs 1A and S1). In contrast, punctate signals were detected near the iRBC membrane and inside the parasite in the binary form, suggesting that SBP3 was expressed and exported at the late stage of intraerythrocytic development. VESA-1 and SBP4 were detected in the parasites in both the single and binary forms. In single-form parasite VESA-1 was detected mainly inside the parasite but not in the cytosol or near the membrane of the iRBCs, whereas SBP4 was detected inside the single-form parasite and also in the cytosol of iRBCs. Colocalization of SBP3 and VESA-1 near the membrane of the iRBC in the binary-form parasites implies ridge localization of SBP3. Colocalization of SBP3 and SBP4 in the binary-form parasites in confined areas, such as specific organelles, suggested their spherical body localization; however, colocalization was detected only in a few areas within the iRBCs (Fig 1A). *Pearson's correlation coefficient (PCC)* value of anti-Myc (SBP3) and VESA-1 fluorescence nearby the surface of iRBC showed 0.72±0.01 (average ± SE) confirmed their colocalization, however that of the anti-Myc (SBP3) and SBP4 showed lower value 0.34±0.02 (average ± SE; Fig 1B). PCC values of fluorescence inside the parasites showed 0.45 ±0.05 and 0.61±0.02 (average ± SE; SBP3 and VESA-1, and SBP3 and SBP4, respectively). Likewise, PCC values inside the iRBC were 0.49±0.02 and 0.29±0.03 (average ± SE; SBP3 and VESA-1, and SBP3 and SBP4, respectively). To confirm the precise localization of SBP3, particularly its association with the ridges, immunoelectron microscopy (IEM) was performed. IEM observation confirmed the expression of SBP3 in spherical bodies of the parasite and also in the ridges of the iRBCs (Figs 1C and S2). As SBP3 does not have a transmembrane domain, the protein may be localized beneath the iRBC membrane at the protrusion. Quantification of the gold particles at the surface of iRBC revealed that SBP3 is significantly located nearby the ridges than other part of the iRBC surface (8.75±1.19 at the ridge and 2.55±0.50 for the other part of the surface, 0 and 0.95±0.30 for those of the negative control, average ± SE; Fig 1D). Moreover, quantification of gold particles inside the parasites confirmed that the most of the particles located at spherical bodies (39.40±6.03, 0.10± 0.07, 0.55±0.21, and 7.35±1.13 for spherical body, nucleus, apical organelle, and parasite cytoplasm, respectively, 0.50±0.19, 0,

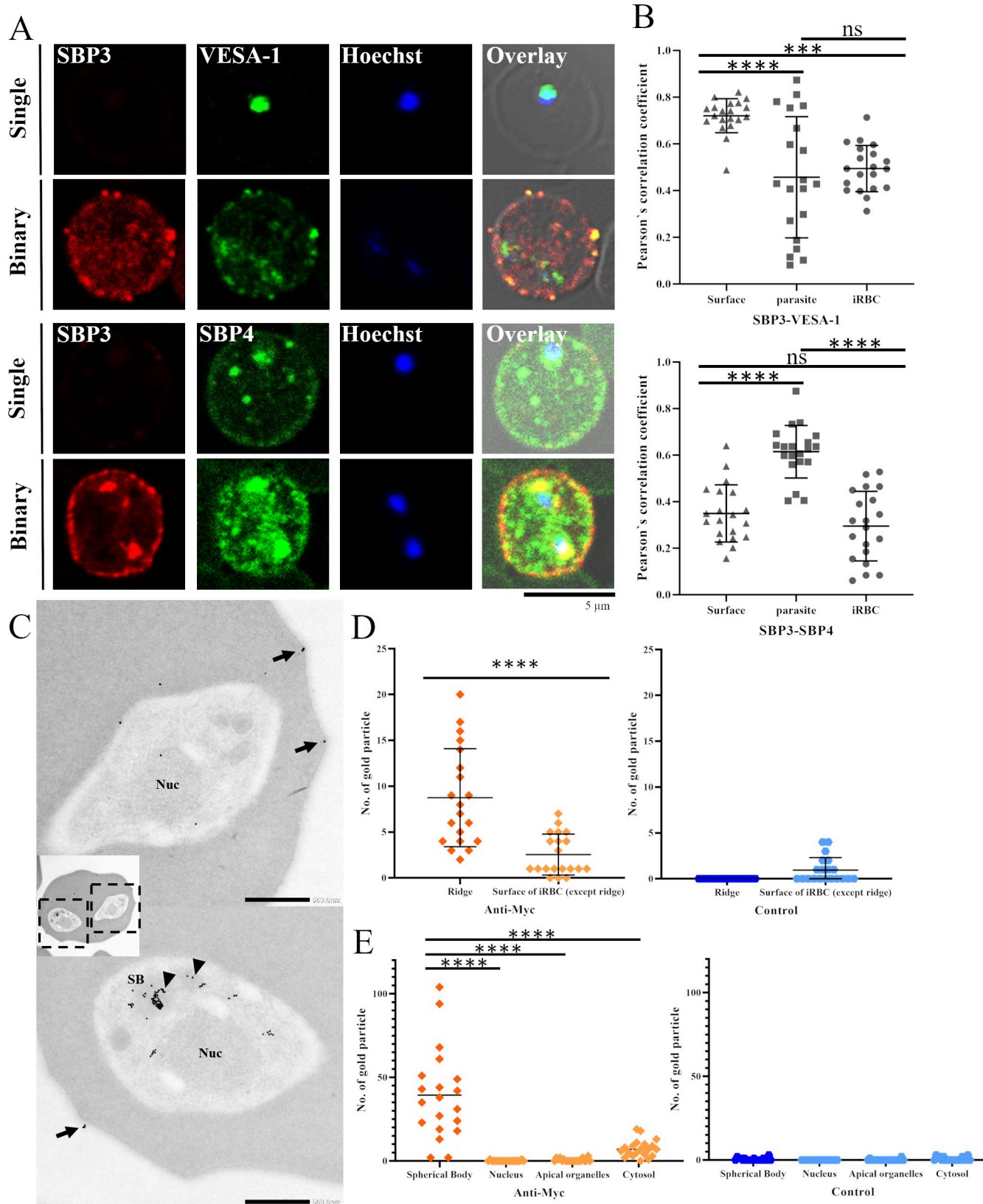

**Fig 1. Localization analysis of SBP3 by indirect immunofluorescence assay (IFA) and immunoelectron microscopy (IEM).** (A) IFA images of *B. bovis* SBP3-Myc parasites. The parasites were stained with anti-Myc antibody (SBP3, red), anti-VESA-1 antibody (green), and anti-SBP4 antibody (green). Nuclei (Nuc) were stained with Hoechst 33342 (Hoechst, blue). Overlay shows bright field, anti-Myc antibody (SBP3, red) and anti-VESA-1 antibody (green) or SBP4 antibody (green). Scale bar = 5 μm. (B) Pearson's correlation coefficient (PCC) values of the fluorescence images. Upper row: SBP3 and VESA-1, lower row: SBP3 and SBP4. Images were analyzed nearby the surface of iRBC, inside the parasites, and iRBC cytosol (n = 20, ns, non-significant; *** $p < 0.001$; **** $p < 0.0001$; determined by multiple comparisons one-way ANOVA test). (C) IEM images of *B. bovis* SBP3-Myc parasites. The parasites were stained with anti-Myc antibody. Gold particles (SBP3) were detected in spherical bodies (SB; arrowheads) and on the iRBC surface ridges (arrows). Scale bar = 500 nm. (D) Quantification of the number of gold particles nearby the ridges (Ridges) and other part of the iRBC surface (Surface of iRBC). Anti-Myc: images reacted with anti-Myc antibody. Control: negative control images reacted with rabbit IgG. (n = 20, n: the number of iRBCs used for counting the gold particles, **** $p < 0.0001$; determined by Mann-Whitney U test) (E) Quantification of the number of gold particles inside the parasites. Gold particles were counted on spherical bodies, nucleus, apical organelles and cytosol of the parasites. (n = 20, **** $p < 0.0001$; determined one-way ANOVA test).

0.16±0.11, and 0.65±0.24 for those of the negative control, average ± SE; Fig 1E). Gold particles were also detected in the iRBC cytoplasm, however, the number was not statistically significant with negative control group (S3 Fig). Altogether, these results revealed that SBP3 and VESA-1 colocalized at the ridges of iRBCs.

## Inducible knockdown reduced SBP3 expression with high efficacy

To elucidate possible functions of SBP3 in *B. bovis*, a *glmS* ribozyme-inducible gene knockdown (KD) system was used. The *glmS* sequence was inserted at the 3′ end of the *sbp3* ORF (open reading frame) along with two Myc-tag sequences. The plasmid containing homologous recombination sites flanking the Myc-*glmS*, *tpx-1* terminator, and *hDHFR* cassette was constructed, and the linearized plasmid was transfected by electroporation into the parasites. Following 10 days of selection by WR99210, the transfectants were obtained and 2 clones (SBP3-*glmS* Tg1 and Tg2) were established (Fig 2A). Successful transfection and integration of the cassette into the desired locus in the parasite genome was confirmed by 3 sets of diagnostic PCRs (Fig 2B). Additionally, whole genome sequencing of the transgenic parasite confirmed that the *glmS* sequence and *hDHFR* cassette were successfully integrated at the desired locus and not in any other places in the genome (S4 Fig).

To evaluate the SBP3 KD efficacy, the two clones of the SBP3-*glmS* parasite were incubated with or without 10 mM of glucosamine (GlcN) for 24 h, according to our previous report for the KD [20]. The total proteins were extracted, and expression of SBP3 was examined by Western blot analysis. A 135-kDa signal corresponding to SBP3 was detected in the absence of the GlcN culture condition. However, the signal was barely visible in the presence of the GlcN culture condition. The comparable results obtained in both clones suggested reproduction of SBP3 KD with high efficacy in the two independent transfectant parasites (Figs 2C and S5A). Quantification of the signal intensities for 3 independent KD experiment followed with Western blot analyses confirmed that the amount of SBP3 was reduced by 74% and 63% in SBP3-*glmS* Tg1 and Tg2, respectively, under GlcN administration (the values were 0.74±0.05, 0.19±0.07, 0.88±0.05, and 0.32±0.16 for Tg1-, Tg1+, Tg2-, and Tg2+, respectively, average ± SE; Fig 2D).

To confirm SBP3 KD was regulated by dose dependent manner, varying concentration of GlcN was administrated to the SBP3-*glmS* parasites. Followed by incubation with 0–10 mM GlcN, Western blot analyses was performed. As a result, all concentrations of GlcN administration resulted in a reduction of SBP3 expression, higher concentration of GlcN reduced SBP3 expression more severely (S5B Fig). In addition, a recovery experiment was performed by KD of SBP3 with 10 mM GlcN for 24 hours and following drug removal. Western blot analysis demonstrated that SBP3 expression was fully recovered after the removal of GlcN (S5C Fig).

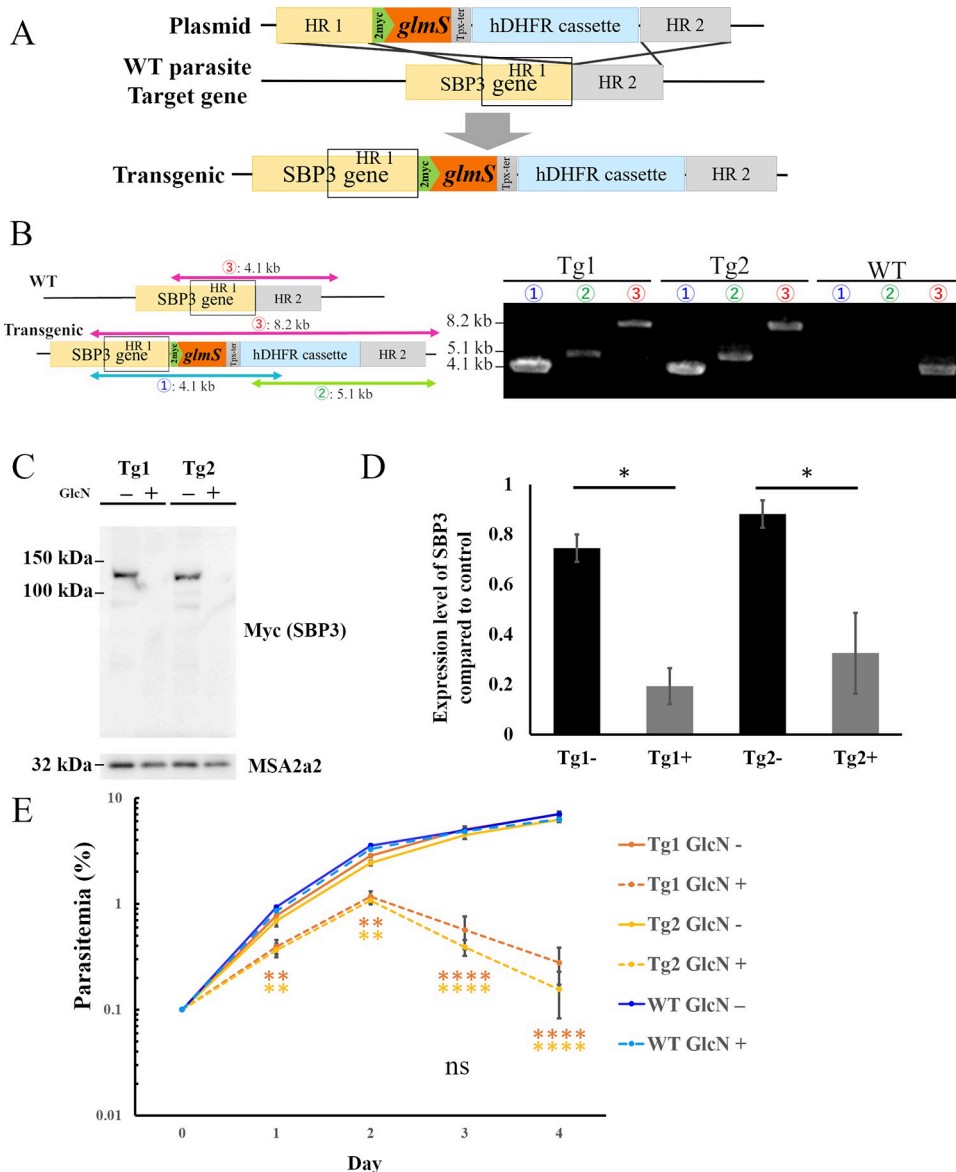

**Fig 2. Generation of *B. bovis* SBP3-inducible knockdown lines using *glmS* system.** (A) Schematic of plasmid to insert Myc-*glmS* sequences at the 3′ end of the *sbp3* ORF showing the two Myc-tag, *glmS*, *thioredoxin peroxidase-1* (*tpx-1*) 3′ noncoding region (NR), *human dihydrofolate reductase* (*hDHFR*) expression cassette, and homologous regions (HR 1 and 2). (B) Agarose gel electrophoresis image of diagnostic PCR to confirm integration of the Myc-*glmS* sequence. Tg1 and Tg2 indicate 2 independent clones. WT: parental strain parasite. (C) Western blot analysis of SBP3-*glmS* clones with (+) or without (-) glucosamine (GlcN). Anti-MSA2a antibody was used to detect MSA2a protein as a loading control. The image is representative of the experiments (other images are shown in S5 Fig). (D) Densitometry of SBP3 protein levels with (+) or without (-) GlcN. The data show average ± SE results of three independent experiments (* p ≤ 0.05; determined by paired Student *t*-test). (E) Growth of WT and SBP3-*glmS* lines with (+) or without (-) GlcN. Initial parasitemia was 0.1%, and parasitemia was monitored for 4 days with daily culture medium replacement. The data are shown as mean ± SE for three independent experiments. (** p < 0.01; **** p < 0.0001; determined by multiple *t*-test).

## SBP3 KD induced growth defect of the parasites

To evaluate the effect of SBP3 KD in intraerythrocytic development of the parasite, growth curves of SBP3-*glmS* Tg1, Tg2 and wild type (WT) parental stain *in vitro* culture in the absence and presence of GlcN were compared. Tg1 and Tg2 grew normally in the absence of GlcN,

whereas their growth in the presence of GlcN was severely impaired. There was no significant difference in growth of WT parasite in the presence or absence of GlcN (the parasitemia at day 4 were 7.01±0.41, 0.27±0.11, 6.25±0.36, 0.16±0.07, 7.03±0.19, and 6.27±0.12 for Tg1 GlcN-, Tg1 GlcN+, Tg2 GlcN-, Tg2 GlcN-, WT GlcN-, and WT GlcN+, respectively, average ± SE; Fig 2E). Comparison of the proportion of single and binary forms of the parasites between the absence and presence of GlcN in the medium did not show a clear difference (S6 Fig). These results suggest that SBP3 has a crucial role in intraerythrocytic development of the parasite.

## SBP3 KD affected iRBC surface localization of VESA-1

To gain further insights into the interplay between VESA-1 and SBP3, the effects of SBP3 KD on the localization and expression of VESA-1 were examined. SBP3-*glmS* clones were double-stained with anti-Myc and anti-VESA-1 antibodies. Immunofluorescence assay (IFA) showed a significant reduction of VESA-1 signals near the surface of the iRBC (intensities were 15587.5±758.8, 4488.7±668.0, 9340.1±1048.4, and 1742.7±346.9 for SBP3-, SBP3+, VESA-1-, and VESA-1+, respectively, average ± SE; Figs 3A, 3B and S7), inside the parasite and cytosol of iRBC on SBP3 KD parasites (S8A Fig), while SBP3 KD did not show a significant reduction of SBP4 signals (intensities of iRBC surface were 44942.6±3832.6, 7734.3±897.3, 24575.1 ±2126.5, and 22250.2±3176.6 for SBP3-, SBP3+, SBP4-, and SBP4+, respectively, average ± SE; Figs 3A, 3B and S8A). However, the overall expression of VESA-1 by Western blot analysis was not changed. Quantification of the signal intensities for 3 independent KD experiment followed by Western blot analysis confirmed that the amount of SBP4 and VESA-1 did not reduced (the values of SBP4 were 0.65±0.17, 0.79±0.32, 0.68±0.20, and 0.79±0.21 for Tg1-, Tg1 +, Tg2-, and Tg2+, respectively, the values of VESA-1 were 0.40±0.04, 0.48±0.05, 0.60±0.12, and 0.62±0.18 for Tg1-, Tg1+, Tg2-, and Tg2+, respectively, average ± SE; Figs 3C, 3D and S9). These data suggest both a notable reduction of VESA-1 on the surface of iRBCs after KD of SBP3 and possible interplay between VESA-1 and SBP3, which facilitates localization of VESA-1 on the iRBC surface but not its expression. In contrast, localization and expression of SBP4 were unchanged by SBP3 KD on IFA with anti-Myc and anti-SBP4 antibodies and the Western blot analysis in SBP3-*glmS* clones (Figs 3A-3D and S9). Additionally, the PCC values showed a significant reduction at the surface of iRBC between SBP3 and VESA-1 on SBP3 KD parasites (S8B Fig). These results suggested an absence of interplay between SBP3 and SBP4 and an assumption that they have different functions.

## SBP3 KD abolished ridge structures on the iRBCs

Structural changes of iRBCs by SBP3 KD were observed by transmission electron microscopy (TEM). As a result, a severe reduction in ridge numbers on the surface of iRBCs was observed (Figs 4A and S10). The ridge numbers counted on the iRBCs of SBP3-*glmS* Tg1 and Tg2 grown in the absence of GlcN were 6.32±0.79 and 5.96±0.41 (average±SE), respectively. However, the numbers decreased dramatically to 0.28±0.17 and 0.04±0.04 for Tg1 and Tg2, respectively, when the parasites were grown in the presence of GlcN under the SBP3 KD condition (Fig 4B). In this experiment, clear morphological changes were not observed in spherical bodies and other organelles of the parasites. Nevertheless, this noticeable phenotype regarding ridge number reduction demonstrated the crucial role of SBP3 in ridge formation.

## SBP3 KD reduced iRBC cytoadhesion to bovine brain endothelial cells

To evaluate the effect of SBP3 KD on the cytoadhesion of iRBCs to BBECs, cytoadhesion assays were conducted. The SBP3-*glmS* Tg1 and Tg2 were treated with or without GlcN for 24 h and further incubated with BBECs, and cytoadhering iRBCs were counted for 100 BBECs. Because

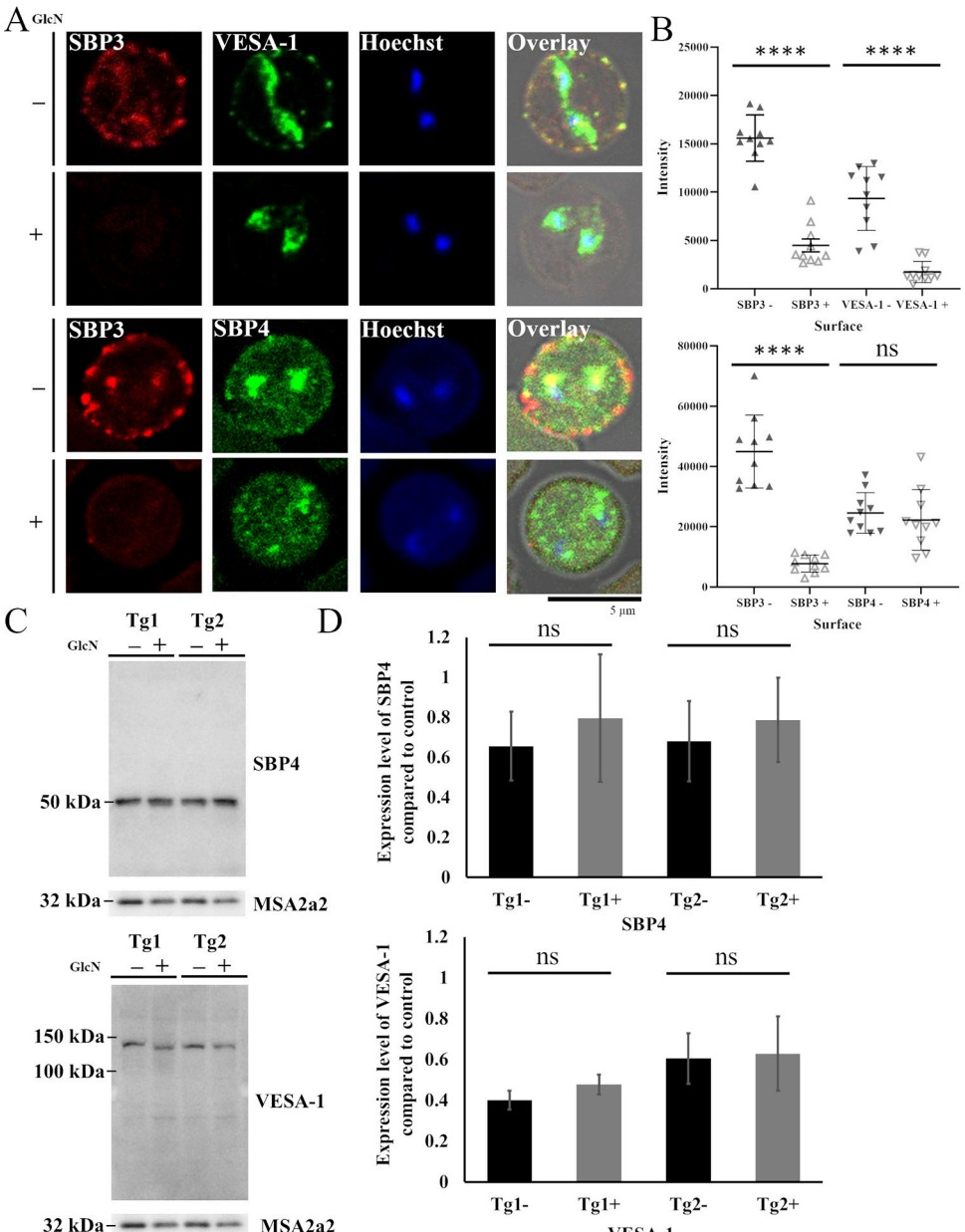

**Fig 3. SBP3 KD decreases the surface localization of VESA-1.** (A) IFA images of SBP3-*glmS* parasite with (+) or without (-) GlcN. Parasites stained by anti-Myc antibody (SBP3, red), anti-VESA-1 antibody (green), anti-SBP4 antibody (green), and nuclei stained with Hoechst 33342 (Hoechst, blue) are shown. Overlay shows bright field and fluorescent images. Scale bar = 5 μm. (B) Fluorescence intensities of SBP3, VESA-1 and SBP4 near the surface of iRBCs before (-) and after (+) SBP3 KD (n = 10, ns; non-significant; **** p < 0.0001; determined by multiple t-test) (C) Western blot analysis of SBP3-*glmS* clones with (+) or without (-) of glucosamine (GlcN). Anti-VESA-1 and Anti-SBP4 antibodies were used to detect the proteins. Anti-MSA2a antibody was used to detect MSA2a protein as a loading control. The image is representative of three independent experiments. (D) Densitometry of SBP4 and VESA-1 protein levels with (+) or without (-) GlcN. The data show average±SE of three independent experiments (ns, non-significant; determined by paired Student t-test).

of the growth reduction observed under the SBP3 KD condition, the iRBCs prepared in the absence of GlcN medium were diluted by normal RBCs to adjust parasitemia before the cytoadhesion assay. As expected from the results showing severe defects in surface localization

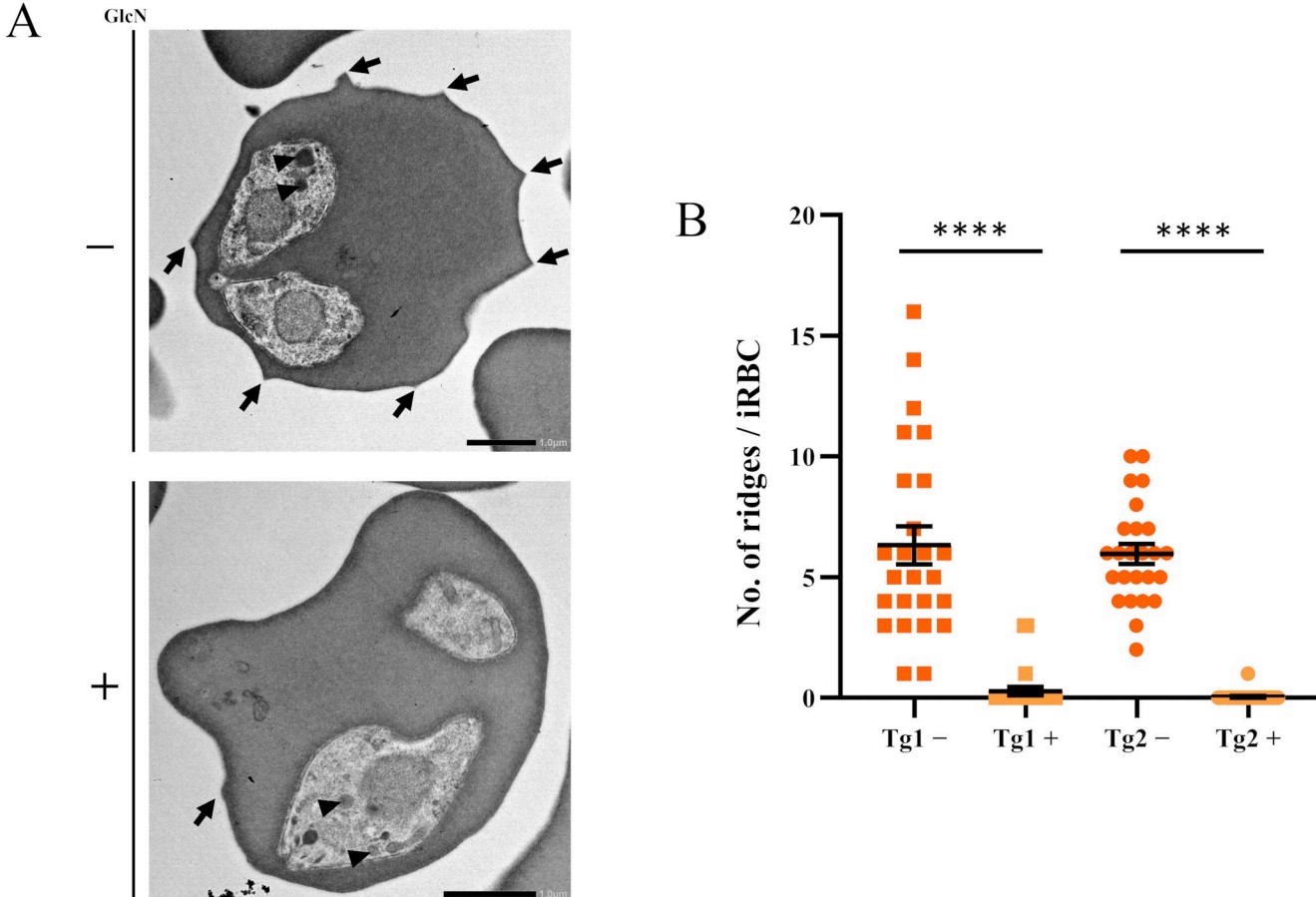

**Fig 4. Severe decrease in ridge numbers on the surface of iRBCs.** (A) Transmission electron microscopy (TEM) images of SBP3-*glmS* parasite iRBCs with (+) or without (-) GlcN. Arrows indicate ridges and arrowheads indicate spherical bodies. Scale bar = 1 μm. (B) Quantification of ridge numbers on the surface of iRBCs of SBP3-*glmS* parasites in the presence or absence of GlcN. Ridge numbers were counted for 25 iRBCs (\*\*\*\* p < 0.0001 as determined by Mann-Whitney U test).

of VESA-1 and ridge formation on the iRBCs, the number of cytoadhering iRBCs was significantly reduced under the SBP3 KD condition (Fig 5A and 5B). The numbers (average ± SE) of cytoadhering iRBCs for Tg1 and Tg2 grown in the absence of GlcN were 219±28 and 209±21, respectively. However, the numbers were markedly decreased to 38±8 and 42±7 for Tg1 and Tg2, respectively, when the parasites were grown in the presence of GlcN under the SBP3 KD condition. There was no significant cytoadhesion ability differences on the parental parasite in the absence or presence of GlcN (225±1 and 221±7 for WT- and WT+, respectively; Fig 5B). Because cytoadhesion of iRBCs to BBECs is one of the main pathologies leading to cerebral babesiosis, this result suggested a strong association of SPB3 to the pathogenicity of the parasite.

## SBP3 interacts with Band 3 multiprotein complex

Lastly, to gain insights into the SBP3 KD phenotype, an RNA-seq analysis and proteomic analysis were performed. Total RNA was extracted before and after GlcN administration and served for RNA-seq analysis. The reduction in *sbp3* transcription in the transcriptome data set confirmed the successful KD (after KD/before KD reads ratio was 0.08, S1 Table). Clear changes in the expression of *sbp1*, *sbp2*, and *sbp4* were not observed (after KD/before KD

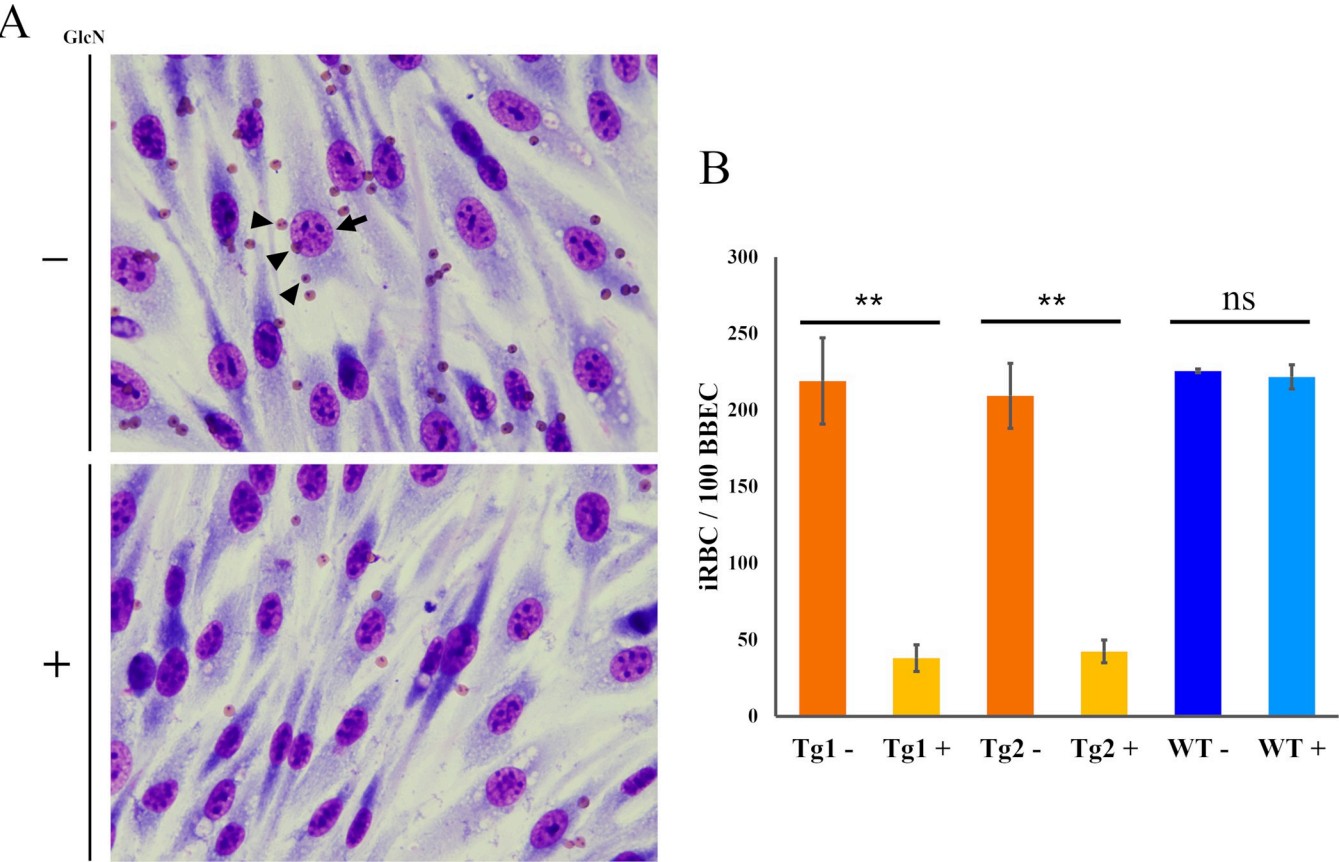

**Fig 5. SBP3 KD decreases cytoadhesion of iRBCs to bovine brain endothelial cells (BBECs).** (A) Cytoadhesion assay of SBP3-*glmS* parasites with (+) or without (-) GlcN. Arrow indicate the BBECs and arrowheads indicate iRBCs. (B) Quantification of cytoadhered SBP3-*glmS* and WT parasite iRBCs to the 100 BBECs. The data are shown as the mean ± SE of a triplicate assay (ns, non-significant; ** p < 0.01 as determined by paired Student *t*-test).

reads ratios were 1.20, 0.98, and 0.93, respectively). Several genes coding for putative integral membrane proteins (e.g., BBOV_IV010380 and BBOV_I003270) and genes coding for rhoptry proteins (rhoptry neck protein RON5: BBOV_IV011430 and rhoptry-associated protein 1 [RAP-1] family protein: BBOV_IV009860) were detected as the genes that reduced expression under KD. Several genes coding for heat shock proteins (Hsp70 family protein: BBO-V_III004240 and putative heat shock protein 90–3: BBOV_III004230) were also found as genes that increased expression under KD. Although further analysis is necessary, this result suggested that while SBP3 KD does not affect the expression of other SBPs, it could potentially affect the expression of membrane proteins, secretory organelle proteins other than spherical bodies, and chaperone proteins.

Proteomic analysis was performed to detect proteins that interact with SBP3. The total protein fractions extracted from the iRBCs of SBP3-Myc were immunoprecipitated with anti-Myc antibody, and the precipitated fractions were analyzed by liquid chromatography-tandem mass spectrometry. Immunoprecipitation using parental *B. bovis* parasites served as a control experiment. When the detected proteins (SBP3 and proteins detected more from the control sample were excluded) were listed based on the spectrum count order, the bovine anion exchange protein (Band 3) was the most detected protein, followed by bovine protein 4.2 and hemoglobin subunit beta (Table 1 as a representative data; full data of three repeat experiments is shown at S2 Table. Similar results were obtained from three experiments). Moreover,

**Table 1. Proteins detected by SBP3 immunoprecipitation.**

| | Proteins | Species | Accession Number | Spectrum | |
|---|---|---|---|---|---|
| | | | | Parental strain | SBP3-Myc |
| 1 | Anion exchange protein (Band 3) | *Bos taurus* | F6QPG2 | 8 | 40 |
| 2 | Protein 4.2 | *Bos taurus* | O46510 | 0 | 20 |
| 3 | Cluster of Hemoglobin subunit beta | *Bos taurus* | P02070 | 1 | 11 |
| 4 | Uncharacterized protein | *Bos taurus* | A0A3Q1NCL2 | 1 | 9 |
| 5 | ANK_REP_REGION domain-containing protein (Ankyrin 1) | *Bos taurus* | F1MY81 | 0 | 9 |
| 6 | Cluster of Histone H4 | *Bos taurus* | A0A3Q1M1Z1 | 5 | 7 |
| 7 | Cluster of Membrane protein, putative (MTM) | *B. bovis* | A7AQB0 | 0 | 7 |
| 8 | Spectrin beta chain | *Bos taurus* | F1MKE9 | 2 | 6 |
| 9 | Cluster of Heat shock 70 kDa protein 1B | *Bos taurus* | Q27965 | 3 | 5 |
| 10 | Cluster of Uncharacterized protein | *Bos taurus* | G3MW09 | 0 | 5 |
| 10 | Uncharacterized protein | *B. bovis* | A7AVY5 | 0 | 5 |
| 10 | Uncharacterized protein | *B. bovis* | A7AV50 | 0 | 5 |

ankyrin 1 was detected as the fifth most detected protein and spectrin beta chain was detected as the eighth. *B. bovis* protein was also detected as the seventh most detected protein, and it was MTM coded by BBOV_III011920. When the MTM-Myc parasite, which was produced in our previous study [20], was stained with anti-Myc and anti-SBP3 antibodies, the fluorescence signal was colocalized near the membrane of the iRBCs, suggesting their association with the ridges (S11 Fig). In contrast, SBP1, 2, and 4, VESA-1, and VEAP were not detected. Band 3 is the most abundant protein in the erythrocyte membrane, forming a complex with other erythrocyte proteins such as ankyrin, band 4.1 and band 4.2 [21–24]. This data suggested that SBP3 has an association with the host Band 3 multiprotein complex during ridge formation on the iRBCs.

## Discussion

SBPs are a group of proteins named based on localization in spherical bodies of *B. bovis* that are exported to the cytosol and the juxtamembrane space of iRBCs. Four SBPs (SBP1-4) have been found in the *B. bovis* genome. SBP1, SBP2t11, SBP3 and SBP4 have a signal peptide and a PEXEL-like motif at the N-terminus, but they do not share clear conserved domains or apparent similarities in sequence [8,9,11,12,25]. Because of the juxtamembrane localization of SBPs in the iRBC, their involvement in ridge formation has been speculated for a long time, but no clear evidence has been presented to date [4]. A recent study indirectly supported the involvement of SBP2 in cerebral babesiosis, in which overexpression of SBP2t11, one of the truncated copies identified in the genome, significantly interfered with the cytoadhesion of iRBCs to BBECs [18].

At the initiation of the present study, several truncated copies of *sbp1* that could have hampered targeting of the desired locus in the transfection experiments were identified in the genome database (S3 Table). There were 5 truncated copies of *sbp1* together with the full-length copy in the genome, and they shared 5' and 3' sequences with high homologies (S12 Fig). Therefore, *sbp3*, which was confirmed to be a single-copy gene in the genome, was focused on in this study to investigate the roles of SBPs in ridge formation.

The precise localization of SBP3 within the iRBCs and the timing of export were previously unclear. Therefore, to elucidate these aspects, co-staining of SBP3 with either VESA-1 or SBP4 was conducted. The expression and export of SBP3 were mostly observed in the iRBCs with a binary form or mature stage of the parasites, whereas VESA-1 and SBP4 expressions were

detected even in the single-form parasites, suggesting that SBP3 might have a distinct function in the late stage of the parasite's development. Moreover, co-localization of SBP3 and VESA-1 near the RBC membrane with punctate signals suggested their co-expression on the ridges. Limited colocalization of SBP3 and SBP4 suggested their functional difference in the iRBCs. IEM confirmed SBP3 expression on the ridges of the iRBCs, implying the role of SBP3 in ridge formation.

To determine the function of SBP3, we tried to knock out SBP3 with the CRISPR-Cas9 genome editing system [19], but no transgenic parasite appeared after drug selection (schematic of the plasmid is shown on S13 Fig). This result suggested that SBP3 may play an important role in parasite survival during the erythrocytic stage, and thereafter we applied the *glmS*-inducible KD system. The KD of SBP3 was confirmed by Western blotting, IFA, and RNA-seq analysis, and all three methods showed that KD was effectively achieved following GlcN administration. The KD of SBP3 significantly affected parasite growth. The parasitemia was slightly increased after the GlcN treatment, possibly due to the remaining SBP3 after the induction of KD, but the parasitemia dropped at day 3. The growth defect phenotype was more severe than that seen in our former KD of VESA1-export associated protein (VEAP) using the *glmS* system [20]. It is not clear why SBP3 knockdown severely affected parasite growth. RNA-seq analysis showed downregulation of the expression of several genes encoded for putative membrane proteins and rhoptry proteins. As discussed in the following sections, our current understanding suggests that abolishment of ridges could lead to the mis-localization of some exported proteins, and possibly by disturbing some export machinery and protein trafficking, it resulted in significant growth defects. However, further analysis is necessary, and the association between SBP3 and rhoptry proteins needs more clarification.

Electron microscopy observation showed a severe decrease in ridge numbers on the surface of iRBCs (schematic diagram is shown in Fig 6A). This phenotype was more obviously observed in SBP3 KD than that observed in our previous VEAP KD experiment [20]. On the contrary, SBP3 KD resulted in decreased cytoadhesion of iRBCs to BBECs, but this phenotype was more prominent in VEAP KD, which almost completely abolished the cytoadhesion [20]. Knockdown of VEAP in our previous study and SBP3 in this study both resulted in the mis-localization of VESA-1 on the surface of iRBCs; however, direct interactions between VEAP and VESA-1 were not observed in our previous study [20]. Because VEAP was expressed only in the cytosol of iRBCs, it was hypothesized that VEAP might play a role in facilitating proteins exportation in the cytosol [20]. In the present study, SBP3 KD affected the localization of VESA-1 on the iRBCs, but Western blotting and RNA-seq analysis suggested that the expression of VESA-1 was not affected. Additionally, VESA-1 was not pulled down by immunoprecipitation as an interaction counterpart of SBP3. These results also suggest that SBP3 and VESA-1 do not interact directly. In addition, expression and localization of SBP4 were not affected by SBP3 KD, and SBP4 was not detected by immunoprecipitation. These results suggested that SBP3 and SBP4 were exported independently from the spherical bodies and that they had separate functions. While the localization of VESA-1 in the parasites remains unclear, SBP3, SBP4, and VEAP are initially localized within the spherical bodies before their export to the cytosol of iRBCs. However, the localization pattern on iRBCs shows similarities between VESA-1 and SBP3, thus distinguishing it from SBP4 and VEAP. Further investigation into the export mechanisms and functions of these proteins is warranted.

Interestingly, our immunoprecipitation experiment revealed several host RBC submembrane cytoskeleton proteins such as band 3, band 4.2, ankyrin 1 and spectrin beta as interaction counterparts of SBP3. Band 3 is a membrane transporter protein crucial for numerous physiological processes, notably influencing the mechano-structural properties of erythrocytes [26,27]. It forms complexes with other proteins within erythrocytes, including cytoskeletal

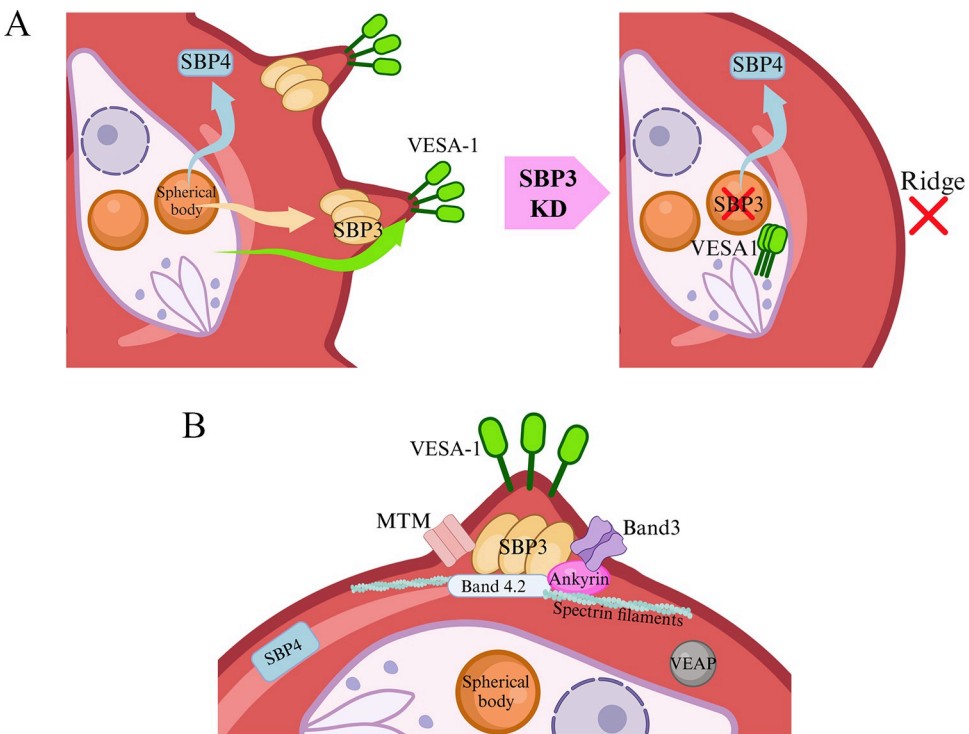

**Fig 6. Schematic diagram of ridge-related proteins in *Babesia* iRBC.** (A) Effect of SBP3 knockdown on ridge formation and localization of SBP3, VESA-1, and SBP4. When SBP3 was knocked down, VESA-1 was not exported to the surface of iRBCs, while there was no change in the expression and localization of SBP4. (B) *B. bovis* exportome localization and interaction of ridge-related proteins with iRBC proteins. SBP3 localizes at the ridge and interacts with iRBC cytoskeleton proteins (Band 3, Band 4.2, ankyrin 1 and spectrin filaments). MTM and VESA-1 are localized on the surface of ridges. VEAP and SBP4 are located in the iRBC cytoplasm. The schematic diagram was created with BioRender.com software.

proteins such as ankyrin, band 4.1, and band 4.2 [21–24]. However, MTM, a parasite protein, was also detected as an interaction counterpart by immunoprecipitation and colocalize with SBP3 near the membrane of iRBC. MTM is a multi-transmembrane protein and is suggested to serve as a channel or transporter for the uptake of nutrition necessary for the parasites [20]. Combined with the phenotype results of the SBP3 KD parasites and the immunoprecipitation data, a crucial role of SBP3 in ridge formation was strongly suggested with the hypothesis that SBP3 binds to the RBC cytoskeleton and changes the structure of the iRBC membrane to form the ridges, providing a place that other exported proteins, including VESA-1 and MTM, can settle on the surface of iRBC (Fig 6B). In addition, SBP3 is a conserved protein among various *Babesia* species [25], including those lacking ridge and cerebral babesiosis, suggesting that SBP3 may have a conserved function across these species.

*Plasmodium falciparum*, a causative agent of human malaria, exhibits symptoms called cerebral malaria, which shares several characteristics with bovine cerebral babesiosis [5,28,29]. Cytoadhesion of iRBCs to human brain endothelial cells is observed in cerebral malaria, and the protrusion on the iRBCs called "knob" is associated with cytoadherence of the iRBCs [30]. PfEMP1 is a *P. falciparum* antigen known to be the major ligand for cytoadhesion [31], but PfEMP1 and *B. bovis* VESA-1 do not share any similarity in amino acid sequence. In *P. falciparum*, it is reported that knockout of knob-associated histidine-rich protein (KAHRP) prevents the appearance of knobs on the surface of the iRBCs and decreases cytoadhesion of the iRBCs to platelets [32,33]. KAHRP interacts with band 4.1, spectrin, actin, and ankyrin of the host

iRBCs and facilitates the rearrangement of the submembrane cytoskeleton to form knob structures [34–38]. It appears that SBP3 may be a functional ortholog of KAHRP even though they do not share any similarity in amino acid sequence other than having a signal peptide. In addition, direct interaction between SBP3 and VESA-1 was not observed in the present study, whereas direct interaction between KAHRP and PfEMP1 has been reported in *P. falciparum* iRBCs [39,40].

In conclusion, this study revealed that SBP3 plays an important role in ridge formation on iRBCs. Furthermore, SBP3 is crucial for intraerythrocytic development of the parasite, localization of VESA-1 to the ridge, and cytoadhesion of iRBCs to BBECs. As surface localization of VESA-1 on ridges and cytoadhesion of the iRBCs to BBECs are important prerequisites for the formation of cerebral babesiosis, this study highlights the significance of SBP3 in the pathogenesis of *B. bovis*. The molecular machinery functioning in export and interaction of SBP3 from spherical bodies and to the cytoskeleton proteins for formation of ridges will require further elucidation. Overall, this study may contribute to the discovery of new therapeutic interventions against *B. bovis*.

## Methods

### Parasite cultivation

*Babesia bovis* Texas strain was maintained in continuous culture in GIT medium (Fujifilm Wako Pure Chemical) with purified bovine RBCs (Japan BioSerum) at 10% hematocrit in a microaerophilic stationary-phase culture system. iRBCs that were cytoadherent to BBECs were selected and cloned in our previous study [20]. The *B. bovis* SBP3-Myc parasite was generated in our former study [19].

### Plasmid construction and *B. bovis* transfection

A plasmid was constructed for *B. bovis* SBP3-*glmS* parasites (Fig 2A). The sequences of two Myc tag, *glmS*, *thioredoxin peroxidase-1* (*tpx-1*) 3′ NR, *B. bovis actin* 5′ NR (*act* 5′ NR), *human dihydrofolate reductase* (*hdhfr*), and *B. bovis rhoptry associated protein* 3′ NR (*rap* 3′ NR) were amplified by PCR from our previously constructed plasmids [20]. The PCR product was cloned into XhoI- and SalI-digested sites of pBluescript SK plasmid using an In-Fusion HD Cloning Kit (Takara Bio Inc.). Afterward, the 5′ and 3′ homologous recombination sites were amplified from *B. bovis* genomic DNA, and each PCR product was inserted into the plasmid at the XhoI and SalI sites, respectively. The primer sets used for this study are shown in S4 Table. For the transfection, 20 μg of linearized plasmid was mixed with Human T Cell Nucleofector Kit solution and transfected using a Nucleofector 2b device (Amaxa Biosystems) as previously described [41,42]. Stable transfectants were selected by 20 nM WR99210 and cloned by limiting dilution. The precise integrations of the *glmS* and *hDHFR* cassette sequences were confirmed by diagnostic PCR using primer pairs Bbsbp3_iKD_transconfirm1_F and Rap_hDHFR_R, 5′Act_hDHFR-F and Bbsbp3_iKD_HR3_seminested_R, and Bbsbp3_iKD_-transconfirm1_F and Bbsbp3_iKD_HR3_seminested_R (S4 Table).

Infected RBCs were lysed with saponin (0.15%), then washed 3 times with PBS. The parasite DNA was extracted by NucleoSpin Blood QuickPure (Macherey-Nagel) and sent to Macrogen Japan for genome analysis. The quality and quantity of the purified DNA was assessed using a Bioanalyzer (Agilent). The library was constructed using the TruSeq DNA PCR-Free Library Prep kits (Illumina), and 150 bp pair-end reads were sequenced using NovaSeq 6000 (Illumina). The obtained reads were aligned with the expected recombinant genome sequence using Bowtie 2 then their sequence depth was examined. The expected recombinant genome was constructed based on genome sequence of the T2Bo strain release-63.

## Glucosamine administration and monitoring of parasitemia

SBP3 KD was induced by 10 mM of GlcN (G1514-100G; Sigma-Aldrich Co.) in GIT medium, the concentration of which was determined in our previous study [20]. The iRBCs were incubated with or without GlcN for 24 hours, and samples were collected for further assays. Parasite growth was monitored daily by microscopic observation of Giemsa-stained thin blood smears. The initial parasitemia was 0.1%, and three independent experiments with triplicate well culture was continued for 4 days with or without GlcN. To determine parasitemias, 2000 erythrocytes were observed.

To analyze the SBP3 KD with varying GlcN concentration, SBP3-*glmS* parasites were incubated with 0, 1, 2.5, 5, and 10 mM or 0, 0.15, 0.6, 2.5, and 10 mM GlcN for 24 hours and served for Western blot analysis. Recovery from SBP3 KD was confirmed by administration of 10 mM GlcN for 24 hours, followed by removal of GlcN. The parasites were incubated until recovery of the parasitemia (72 hours) and served for Western blot analysis.

## Immunofluorescence assay

Thin blood smears were prepared from the *in vitro* cultured parasites and fixed with an acetone:methanol mixture at the ratio of 1:1 at -30°C for 10 min, then incubated in PBS-0.05% Tween 20 (v/v) containing 10% normal goat serum for 30 min. Mouse anti-Myc monoclonal antibody (9B11, 1:500 dilution; Cell Signaling Technology) and rabbit antisera against VESA-1a peptide (1:20) or rabbit antisera against SBP4 (1:200) were reacted at 37°C for 1 h [12,20]. Then, Alexa Fluor 488- or 594-conjugated goat anti-mouse IgG and Alexa Fluor 488- or 594-conjugated goat anti-rabbit IgG (1:200; Thermo Fisher Scientific) were reacted at 37°C for 1 h. These antibodies were diluted using PBS-Tween containing 1% normal goat serum, and the parasite nuclei were stained with Hoechst 33342 (1:200). Images were obtained using a confocal laser scanning microscope (TCS-SP5; Leica Microsystems).

## Image analysis

The Pearson's correlation coefficient (PCC) values were analyzed for 20 individual images on SBP3-Myc parasites. Similarly, 10 images were analyzed on SBP3-*glms* parasite before and after SBP3 KD. PCC values were quantified using ImageJ Fiji (https://imagej.net/software/fiji/downloads) with BIOP JACoP plugin between SBP3 and VESA-1 or SBP4. Regions of interest (near the surface of iRBC, inside the parasite, and iRBC cytosol) were set manually based on the bright field images and PCC values were measured.

## Western blotting

Infected RBCs were lysed with saponin (0.15%) in PBS and washed 3 times with protease inhibitor cocktail solution (PI, cOmplete Mini; Merck). Proteins were extracted using 1% Triton-X 100 (w/v) and 2% SDS at 4°C followed by freezing and thawing 3 times. Proteins were separated by polyacrylamide gel electrophoresis (5–20% gradient gel; ATTO Co.) and transferred onto polyvinylidene fluoride membranes (Clear Blot membrane P; ATTO Co.). The membrane was blocked with Blocking One (Nacalai Tesque, Inc.) for 90 min and then reacted overnight with mouse anti-Myc monoclonal antibody (9B11, Cell Signaling Technology) (1:1000), or rabbit anti-VESA-1a peptide antisera (1:100), or rabbit antisera against SBP4 (1:500) [12,20]. In parallel, an equal volume of proteins served as a loading control, and rabbit anti-MSA2a peptide antisera (antisera against peptide CSPQGPTAESPSQADHPTK; 1:500 dilution) was reacted. As a secondary antibody, horseradish peroxidase-conjugated anti-mouse IgG or anti-rabbit IgG (1:25000; Promega) was reacted for 3 h. Bands were detected

with Immobilon Western Chemiluminescent HRP substrate (Millipore), and images were captured by a chemiluminescence detection system (LuminoGraph II EM; ATTO Co.). The band intensities were analyzed by CS Analyzer software (ATTO Co.).

## Electron microscopy and immune electron microscopy

The electron microscopy observations were performed as previously described [20]. In brief, TEM samples were fixed with 2% glutaraldehyde in 0.1 M sodium cacodylate buffer and post-fixed with 1% $OsO_4$ (Nacalai Tesque). The samples were washed, dehydrated in a series of ethanol and acetone washes, and embedded in Quetol 651 epoxy resin (Nisshin EM). Ultra-thin sections were stained and examined at 80 kV under a transmission electron microscope (JEM-1400Flash; JEOL). Samples for IEM were fixed with 4% paraformaldehyde and 0.1% glutaraldehyde in 0.1 M phosphate buffer, followed by dehydration with a series of ethanol washes and embedding in LR White resin (Electron Microscopy Sciences). Thin sections were blocked with 5% non-fat milk (Becton, Dickinson and Company) and 0.0001% Tween 20 in PBS at 37°C for 30 min and incubated with rabbit anti-Myc polyclonal antibody (1:100; ab9106, Abcam) or control normal rabbit IgG at 4°C overnight. After washing, the sections were incubated with goat anti-rabbit IgG conjugated to 15-nm gold particles (1:20; EY Laboratories Inc.), and fixed with 0.5% $OsO_4$ at room temperature for 5 min. The sections were stained and observed by TEM.

## Cytoadhesion assay

Cytoadhesion assay was done as described previously [16]. BBECs (Cell Applications Inc.) were cultivated in 12-well plates with $18 \times 18$-mm cover glasses (Matsunami Glass). The *B. bovis* iRBCs with or without GlcN treatment were adjusted to 1.5% parasitemia with 1% hematocrit and co-cultivated with BBECs for 90 min. The culture plate was gently agitated every 15 min. The wells were washed 3 times with Hanks' balanced salt solution to remove the nonadherent iRBCs, then fixed with methanol and stained with Giemsa solution. The number of adhered iRBCs was counted for 100 BBECs.

## Immunoprecipitation and liquid chromatography-tandem mass spectrometry (LC-MS/MS) analysis

*Babesia bovis* SBP3-Myc and its parental wild-type parasites were used for immunoprecipitation. The iRBCs were initially treated with 0.2% saponin on ice for 20 min to remove hemoglobin, and parasite pellets were suspended with PBS containing 1 mM EDTA and PI. Immunoprecipitations were performed using membrane-permeable chemical crosslinker DSP (Sigma) as described previously [43]. Parasite pellets were treated with 1 mM DSP dissolved in dimethyl sulfoxide for 15 min on ice. Parasite pellets were solubilized with 1% (v/v) Triton X-100 in Tris-buffered saline (TBS)-based solubilization buffer containing 1 mM EDTA, 10% glycerol, and PI at 4°C for 1 h. The protein extract was precleared by incubation with nProtein A-Sepharose 4 Fast Flow (GE Healthcare) for 30 min and then incubated with 20 uL of rabbit anti-Myc antibody (ab9106, Abcam) for 4 h at room temperature with gentle rotation. The protein fractions were mixed with 20 μL of a 50% suspension of nProtein A-Sepharose 4 Fast Flow for an additional 2 h at room temperature. All immunoprecipitation reactions were washed 4 times with 0.5% Triton X-100 in TBS-based solubilization buffer. The beads were then mixed with c-Myc peptide (Thermo Fisher Scientific) (2.4 μg/μL) dissolved in 0.5% Triton X-100 in TBS-based solubilization and incubated with gentle rotation at 4°C for 12 h. The beads were centrifuged, and the supernatants were collected as an immunoprecipitated fraction. Precleared supernatants were used in immunoblots as input protein fractions.

The immunoprecipitation fractions were subjected to LC-M/MS as described previously [44]. The samples were briefly electrophoresed on SDS-polyacrylamide gel, and the gel containing proteins was excised, fixed with acetic acid/methanol solution, and subjected to LC–MS/MS analysis at the W. M. Keck Biomedical Mass Spectrometry Laboratory, University of Virginia, USA, as previously described [20]. The data analysis was performed by database searching using the Sequest search algorithm against *Bos taurus* and *B. bovis*. Filtering and extraction of data were performed using Scaffold version 4.11.1 (Proteome Software Inc.). Protein identifications were accepted if they could be detected at greater than 90% probability and contained at least 2 identified peptides. The normalized total spectra (quantitative value) of each detected protein from SBP-Myc parasites were compared with that of the parental wild-type parasite. The proteins detected with higher quantitative values from SBP3-Myc parasites than that from parental parasites were considered as candidate interacting proteins.

### RNA extraction and RNA-seq analysis

Total RNA was extracted from *B. bovis* SBP3-Myc parasites with or without GlcN treatment. The iRBCs were lysed with 0.05% saponin and washed 3 times with PBS. TRIzol Reagent (Thermo Fisher Scientific) was added to the samples, and RNA was extracted according to the manufacturer's instructions. The quantity and quality of the purified RNA were confirmed using NanoDrop (Thermo Fisher Scientific) and agarose gel electrophoresis and then sent to Macrogen Japan for analysis. The RNA quality was confirmed by Bioanalyzer (Agilent), the library was constructed using a TruSeq standard mRNA LT Sample Prep kit (Illumina), and 100-bp paired-end reads were sequenced using a NovaSeq 6000 system (Illumina). The obtained reads were aligned to the genome sequence of T2Bo strain release-63 retrieved from PiroplasmaDB [45] with TopHat2 v2.1.1 [46], and the number of reads aligned to each of the coding regions was counted using HTSeq2.0.2 [47].

### Statistical analyses

The PCC values were analyzed using GraphPad Prism10 (MDF Co.) and evaluated using one-way ANOVA test and multiple *t*-test. The number of gold particles on IEM images and ridge numbers on the surface of iRBCs were compared using the Mann-Whitney U test and one-way ANOVA test. The parasitemia, band intensities of Western blotting analysis, single/binary ratio and signal intensities of IFA was evaluated using a multiple *t*-test. The number of cytoadhered iRBCs per 100 BBECs was evaluated using a two-tailed paired Student *t*-test. The values were considered significant if the p-value was below 0.05.

### Supporting information

**S1 Fig. Localization analysis of SBP3 by indirect immunofluorescence assay (IFA).** (A) Negative control images of *B. bovis* SBP3-Myc parasites. The parasites were reacted only with secondary antibodies. Nuclei were stained with Hoechst 33342 (Hoechst, blue). Overlay shows bright field and fluorescent images. Scale bar = 5 μm. (B) Confirmation of stainability of anti-Myc antibody and anti-SBP3 peptide antibody by IFA.
(TIF)

**S2 Fig. Localization analysis of SBP3 by immunoelectron microscopy (IEM).** IEM images of *B. bovis* SBP3-Myc parasites. The parasites were reacted with anti-Myc antibody. Control: negative control images without reaction with anti-Myc antibody. Scale bar = 500 nm.
(TIF)

**S3 Fig. Quantification of gold particles on the iRBC cytosol.** The gold particles were counted for 20 individual images. Anti-Myc: the parasites were reacted with anti-Myc antibody. Control: negative control images without reacting with anti-Myc antibody. The number were 7.1 ±1.02 and 4.1±0.53 (average ± SE; ns, non-significant; determined by Mann-Whitney U test).
(TIF)

**S4 Fig. Whole genome sequencing of the transgenic parasites.** The whole genome of the two clones (Tg1, Tg2) were sequenced by Illumina sequencer. The obtained reads were aligned with the expected recombinant genome sequence, then their sequence depth was examined.
(TIF)

**S5 Fig. Western blot analyses of SBP3-*glmS* KD, dose dependency and recovery experiment.** (A) Analysis of SBP3-*glmS* clones with (+) or without (-) glucosamine (GlcN), Anti-MSA2a antibody was used to detect MSA2a protein as a loading control. The images are other two independent experiments. (B) Western blot analysis of dose dependent GlcN administration on SBP3-*glmS* parasite. Two experiments were performed administrating 0, 1, 2.5, 5 and 10, or 0, 0.15, 0.6, 2.5 and 10 mM GlcN. (C) KD recovery experiment of SBP3-*glmS* parasite. +: SBP3 KD with 10 mM GlcN, R: recovery of SBP3 expression after removing the GlcN.
(TIF)

**S6 Fig. Single-binary ratio of SBP3-*glms* parasite in the absence (-) and presence (+) of GlcN after 24 hours.** The data shows 3 replicates of two clones, 100 iRBCs were counted (Average±SE). Tg1 showed 34.0±0.5 single/ 66.0±0.5 binary parasites before KD and 34.7±1.3 single/ 65.3±1.3 binary parasites after KD. Tg2 showed 37.3±0.8 single/ 62.7±0.8 binary parasits before KD and 38.7±0.3 single/ 61.3±0.3 binary parasites after KD.
(TIF)

**S7 Fig. Negative control images for localization analysis of SBP3 by IFA.** IFA images of *B. bovis* SBP3-*glms* parasites. The parasites were stained only with secondary anti-bodies. Nuclei were stained with Hoechst 33342 (Hoechst, blue). Overlay shows bright field and fluorescent images. Scale bar = 5 μm.
(TIF)

**S8 Fig. Quantification of fluorescence intensity and colocalization values for SBP3-*glms* parasites before (-) and after (+) KD.** (A) Fluorescence intensity of VESA-1 and SBP4 before and after SBP3 KD on the parasite and cytosol of iRBC for 10 individual images (ns, non-significant; **** $p < 0.0001$; determined by multiple t-test). (B) Quantification of colocalization between SBP3 and VESA-1 calculated with Pearson's correlation coefficients for 10 individual IFA images on SBP3-*glms* parasites before (-) and after (+) KD (** $p < 0.01$;**** $p < 0.0001$; determined by multiple t-test).
(TIF)

**S9 Fig. Western blot analysis of SBP3-*glmS* clones with (+) or without (-) of glucosamine (GlcN).** (A) Western blot analysis of SBP3-*glmS* parasites (Tg1, Tg2) with (+) or without (-) of glucosamine (GlcN). VESA-1: protein was detected by anti-VESA-1 antibody. Anti-MSA2a antibody was used to detect MSA2a protein as a loading control. The image is other two independent experiments. (B) SBP4: protein was detected by anti-SBP4 antibody. Anti-MSA2a: loading control. The image is other two independent experiments.
(TIF)

**S10 Fig. Severe decrease in ridge numbers on the surface of iRBCs.** Transmission electron microscopy (TEM) images of SBP3-*glmS* parasite iRBCs before (-) and after (+) KD for two

clones (Tg1 and Tg2). Scale bar = 1 μm.
(TIF)

**S11 Fig. Co-Localization analysis of SBP3 and MTM by indirect immunofluorescence assay.** IFA images using *B. bovis* MTM-Myc parasites [20]. The parasite was stained with anti-SBP3 peptide antibody (SBP3, Red) and anti-Myc antibody (MTM, Green). Nuclei were stained with Hoechst 33342 (Hoechst, blue). Overlay shows bright field and fluorescent images. Scale bar = 5 μm.
(TIF)

**S12 Fig. Alignment of amino acid sequences of SBP1 and truncated copies.**
(TIF)

**S13 Fig. Schematic of plasmid to knockout the SBP3 gene.** The plasmid was constructed based on Hakimi et al., 2019 [19]. The primers to amplify HR1, GFP cassette, and HR2, and insert guide RNA were shown on S4 Table.
(TIF)

**S1 Table. RNA-seq data of the SBP3 KD parasite.**
(XLSX)

**S2 Table. Mass spectrometry data of SBP3 immunoprecipitation.**
(XLSX)

**S3 Table. SBP1 truncated copies.**
(TIF)

**S4 Table. List of primers used in this study.**
(TIF)

## Acknowledgments

This work was conducted at the Joint Usage/Research Centers: National Research Center for Protozoan Diseases, Obihiro University of Agriculture and Veterinary Medicine, Hokkaido, Japan; Institute of Tropical Medicine (NEKKEN), Nagasaki University, Nagasaki, Japan; and the International Institute for Zoonosis Control, Hokkaido University, Hokkaido, Japan.

## Author Contributions

**Conceptualization:** Atefeh Fathi, Hassan Hakimi, Masahito Asada.

**Data curation:** Masahito Asada.

**Formal analysis:** Atefeh Fathi, Masahito Asada.

**Funding acquisition:** Hassan Hakimi, Masahito Asada.

**Investigation:** Atefeh Fathi, Hassan Hakimi, Miako Sakaguchi, Junya Yamagishi.

**Methodology:** Atefeh Fathi, Hassan Hakimi, Miako Sakaguchi, Junya Yamagishi, Masahito Asada.

**Project administration:** Masahito Asada.

**Resources:** Hassan Hakimi, Shin-ichiro Kawazu, Masahito Asada.

**Supervision:** Masahito Asada.

**Validation:** Atefeh Fathi, Masahito Asada.

**Visualization:** Atefeh Fathi, Miako Sakaguchi.

**Writing – original draft:** Atefeh Fathi, Masahito Asada.

**Writing – review & editing:** Atefeh Fathi, Hassan Hakimi, Miako Sakaguchi, Junya Yama-
gishi, Shin-ichiro Kawazu, Masahito Asada.

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
