## [Decision Letter · Decision Letter 0]

17 Jun 2024

Dear Dr. Asada,

Thank you very much for submitting your manuscript "Critical role of Babesia bovis spherical body protein 3 in ridge formation on infected red blood cells" for consideration at PLOS Pathogens. As with all papers reviewed by the journal, your manuscript was reviewed by members of the editorial board and by several independent reviewers. In light of the reviews (below this email), we would like to invite the resubmission of a significantly-revised version that takes into account the reviewers' comments.

Both reviewers noted a lack of a control for the glucosamine treatment for inducing the knockdown of SBP3. The concentration of glucosamine in these experiments is very high and could be responsible for the growth inhibition that you observed. Including the proper control would rule out this possibility and is required to confidently interpret the results. Complementation (as noted by reviewer 1) would also greatly improve this important part of the paper. Both reviewers also requested quantification of protein localization and cytoadhesion assays. This will also be required for the manuscript to be considered for publication. 

We cannot make any decision about publication until we have seen the revised manuscript and your response to the reviewers' comments. Your revised manuscript is also likely to be sent to reviewers for further evaluation.

Sincerely,

Kirk W. Deitsch

Academic Editor

PLOS Pathogens

James Collins III

Section Editor

PLOS Pathogens

Michael Malim

Editor-in-Chief

PLOS Pathogens

orcid.org/0000-0002-7699-2064

Reviewer's Responses to Questions

**Part I - Summary**

Reviewer #1: Cerebral babesiosis in cattle, caused by Babesia bovis, is the most lethal form of this disease, and has many parallels with cerebral malaria in humans. Cerebral disease is associated with cytoadhesion in the brain vasculature via parasite proteins aligned on ridges created in the IRBC membrane. In this manuscript, the authors provide identification of a candidate protein, SBP3, putatively facilitating formation of the ridges. The creation of glmS-tagged SBP3 parasites enabled localization and knockdown of the protein to ascertain phenotypes, and immunoprecipitation for identification of interacting partners. The manuscript is overall well-written with suitable English usage, and the figures are of adequate quality technically. This manuscript has the potential to provide useful information to help understand the biology of this parasite and its pathogenesis, including putative contributions of SBP3. However, there are several shortcomings that affect the rigor and veracity of several major claims from this study, and in the opinion of this reviewer require improvement before being suitable for publication.

Reviewer #2: Like Plasmodium, Babesia parasites replicate inside mammalian RBCs and deploy cytoadherence structures on the infected RBC surface to adhere to vascular endothelium. These parasites modify their RBC hosts by exporting parasite proteins into the RBC to perform these various functions. The function of once such Babesia protein SBP3, was studied here. SBP3 appears to colocalise with cytoadherence protein VESA1 at ridges on the RBC surface. SBP3 was appended with a glmS genetic knockdown switch which when activated reduced protein expression by 80%. This reduced the number of RBC ridges and cytoadherance. SBP3 knockdown also reduced parasite growth which was surprising given that most cytoadherance proteins in the comparatively well-studied Plasmodium system are not essential.

**Part II – Major Issues: Key Experiments Required for Acceptance**

Reviewer #1: [1] To meet modern standards of rigor formal quantification of co-localization needs to be performed on multiple cells to demonstrate statistically that SBP3 and VESA1 co-localize. It is highly subjective and unconvincing from the overlays alone, especially as these are likely best-available images. Statistical quantification of gold particles in different cellular compartments (on a per-area basis) needs to be performed to demonstrate specific association with spherical bodies. In addition to spherical bodies and the IRBC membrane, considerable gold is observed over non-membrane-bound compartments, the nucleus, IRBC cytoplasm, etc. Also, why is gold found on the external surface of the IRBC membrane (lower panel), too far away to be accounted for by antibody length? Again, distributions of gold are being presented in a very subjective manner that is only partially consistent with the authors’ interpretation and do not meet modern standards of rigor (but potentially could be made to do so).

[2] Figure 2 Panel E lacks a wild-type control to demonstrate lack (or presence) of toxicity from long-term treatment with 10 micromolar glcN, an effect that is obscured by the use of a log10 plot. In this reviewer’s experience this is quite a high concentration and can have serious toxic effects, especially after extended exposure, of which there is a common awareness in the malaria community. Without including this control in these experiments the “essentiality” of SBP3 based upon this knockdown experiment is not convincing.

[3] The authors did not attempt to complement the lack of functional SBP3 by simultaneous expression of an unmodified sbp3 gene, a standard approach in genetics studies. Such over-expression should theoretically rescue the cytoadhesive phenotype in the presence of glcN and perhaps even increase it in the absence of glcN. These deficits are critical to the authors’ conclusions regarding SBP3 function in ridge formation, cytoadhesion, and potential pathology.

Reviewer #2: 1. In Plasmodium, 2-3 mM glucosamine is usually the maximum concentration used because it is mildly toxic to the parasites. Here 10 mM is used so there should be a control for this against wildtype parasite to check for toxicity in Fig 2E. Parasite growth was also quantified by counting smears and ideally the counters should be blinded to the identity of the parasite treatments. This applies to all counting in Figs 4 and 5.

2. One of the most unexpected results was that the Babesia parasite grew poorly after GlcN treatment which contrasts with Plasmodium where cytoadherence proteins are usually non-essential. Was any attempt made to investigate why the SBP3 knockdowns were becoming sick? Does SBP3 bind to any RBC or parasite protein that could explain its essential function?

**Part III – Minor Issues: Editorial and Data Presentation Modifications**

Reviewer #1: A. Major concerns

1. Figure 1. There are multiple problems with this figure. (i) It is unnecessary (and unnecessarily confusing) to label SBP3 red in one part of the figure and green in the next, especially when mouse anti-Myc was used to label SBP3 and rabbit antibodies for VESA1 and SBP4. Please use consistent pseudocoloring of specific antigens in images. Please label as “SBP3” rather than as “Myc”. (ii) Why were nuclear images not blended with the antigen images in the overlays? This would aid in assessing antigen localizations. (iii) What does “GlcN” refer to in this figure? Please remove. (iv) The brightfield images are too dark to contribute any information; please adjust. (v) To meet modern standards of rigor formal quantification of co-localization needs to be performed on multiple cells to demonstrate statistically that SBP3 and VESA1 co-localize. It is highly subjective and unconvincing from the overlays alone, especially as these are likely best-available images. (vi) Statistical quantification of gold particles in different cellular compartments needs to be performed to demonstrate specific association with spherical bodies. In addition to spherical bodies and the IRBC membrane, considerable gold is observed over non-membrane-bound compartments, the nucleus, IRBC cytoplasm, etc. Also, why is gold found on the external surface of the IRBC membrane (lower panel), too far away to be accounted for by antibody length? Again, distributions of gold are being presented in a very subjective manner that is only partially consistent with the authors’ interpretation and do not meet modern standards of rigor (but potentially could be made to do so). (vii) Since rabbit anti-SBP3 antibodies were available (Figure S1), please demonstrate co-localization of SBP3 and Myc tag, using rabbit anti-SBP3 and mouse anti-Myc (wild-type parasites could provide neg. control for Myc). This would help to validate localization of SBP3 and data obtained with the Myc tag; confirmation should be included in the supplement. (viii) Please include in the supplement images of negative controls for IFA and IEM experiments.

2. Figure 2. (i) While the diagnostic PCR results support the desired outcome, a second method is needed to confirm lack of ectopic recombination elsewhere in the genome that may not be detectable by this method. The second method could be anything from qPCR of gene copies to full genome sequencing, but diagnostic PCR alone only detects the expected outcome and no longer meets standards of rigor. (ii) Please explain what was used in panel D as “control”? If Tg1- and Tg2- were used their values should be 1.0. If values were normalized against MSA2a2 or something else, please elaborate. (iii) Panel E lacks a wild-type control to demonstrate lack (or presence) of toxicity from long-term treatment with 10 micromolar glcN, an effect that is obscured by the use of a log10 plot. In this reviewer’s experience this is quite a high concentration and can have serious toxic effects, especially after extended exposure, of which there is a common awareness in the malaria community. Without this control the “essentiality” of SBP3 based upon this knockdown experiment is not convincing.

3. Table S1. How many biological replicates were included? Statistical analysis is lacking.

4. Figure 3. (i) See comments about pseudocoloring, labeling, and co-localization analysis in Figure 1; the same issues exist here. (ii) Before claiming “significant reduction of VESA1 signals” please provide actual quantification of signal rather than single images that may or may not be representative. This is an important claim and needs full substantiation. (iii) Please provide band quantification for SBP4 and VESA1 in supplement. (iv) Why is there an apparent change in the size of VESA1 in the Tg1- sample? In that region of the gel this would correspond to 10-15 kDa and possibly a different antigenic variant altogether. How does this affect interpretation of the data?

5. Figure 5. These data are quite clear-cut as far as they go. However, (i) no wild-type parasites were included as a negative control on knockdown/ loss of cytoadhesiveness. Given the number of genes whose regulation was significantly altered by glcN treatment (including up-regulation of BBOV_II005480, an AP2 domain-containing protein and possible transcription factor), and may be affected even in the absence of a glmS tag, it is an overinterpretation to conclude that SBP3 specifically is important to this process. (ii) The authors did not attempt to complement the lack of functional SBP3 by simultaneous expression of an unmodified sbp3 gene, a standard approach in genetics studies. Such over-expression should theoretically rescue the cytoadhesive phenotype in the presence of glcN and perhaps even increase it in the absence of glcN. These deficits are critical to the authors’ conclusions regarding SBP3 function in ridge formation, cytoadhesion, and potential pathology.

Minor issues

6. line 62 (and throughout). “functional ortholog” is an inappropriate and meaningless term. SBP3 and KAHRP are not identical in function, and orthologs are defined by their genetic and evolutionary relationships (Tatusov, R.L. et al. 1997. Science 278: 631), not function specifically. “Functional analogs” or “proteins of analogous function” would be terms much more appropriate to this situation.

7. lines 79-82. This wording makes no sense. Please rewrite for clarity.

8. line 445. Please provide a reference for the anti-VESA1a peptide antibody that demonstrates its universal reactivity with different variant forms of VESA1a.

9. line 203. Before concluding lack of interaction of SBP4 and VESA1, and different functions for SBP3 and SBP4 did the authors attempt co-localization of SBP4 and VESA1? Currently, this feels like too great an extrapolation of existing co-localization data, particularly with the availability of reagents.

10. Figure 4. These are very nice data, but please describe how you defined/identified individual ridges non-subjectively for quantification.

11. line 246 (and throughout). It is likely an overstatement that cytoadhesion of IRBCs is “the main pathology leading to cerebral babesiosis”, and until better substantiated this stance should be softened. (See conflicting outcomes in this regard in Sondgeroth, K.S. et al. 2013. Parasites & Vectors 6: 181; Nevils, M.A. et al. 2000. Parasitol. Res. 86: 437; Canto, G.J. et al. 2006. NY Acad. Sci. 108: 397). The genesis and involvement of cytokine dysregulation in this form of pathogenesis has not been well studied in bovine babesiosis but may be quite important.

12. Table 1. Given that proteins were cross-linked in the approach used (rendering this effectively a proximity-labeling study rather than co-precipitation per se), and the reproducibility difficulties of cross-linking studies, it is important to know the number of biological replicates performed in this experiment. How consistent was the capture pattern? Also, please explain the meaning of the numbers given for each co-captured protein (number of fragments? fmolar levels?).

13. line 304. Figure S2 shows an alignment of translated sequences, not 5’ and 3’ untranslated sequences. Please correct.

14. line 319. Unless attempted many times, failure to recover genetically modified parasites cannot be taken as evidence of essentiality. Please soften the wording.

15. lines 322-327. The essentiality of SBP3 based upon the KD data is unconvincing for the reasons discussed earlier, even though it may be true. Please soften the wording.

16. lines 329-333. Other than loss of ridges, were any morphologic alterations observed in KD parasites?

17. line 339. “Knockdown of both..” is ambiguous. Please be specific which proteins were knocked down, and were they knocked down simultaneously?

18. line 396. While it is obvious that SBP3 KD affects ridge development it does not prove a vital role for SBP3 in ridge development. This would require additional experimentation. Rather, the results are consistent with it playing such a role and make it an attractive candidate to play such a role.

19. line 502. Please define “PI”.

20. line 542. Please describe how “band intensities” were quantified.

Reviewer #2: 3. Is there some way of quantifying the localisation of SBP3 with VESA and SBP4 perhaps by using Pearson’s coefficients for 20-30 cells? There needs to be some quantification.

4. In Fig 2A, I can’t tell which of the integration PCR products corresponds to which gel lanes. The primers pairs need to be indicated on the diagram.

5. For the cytoadherence analysis following SBP3 knockdown, were the people counting the slides blinded to the identity of the parasite treatments?

6. Line 270. Chaperone.

PLOS authors have the option to publish the peer review history of their article (what does this mean?). If published, this will include your full peer review and any attached files.

Reviewer #1: No

Reviewer #2: No
---

## [Decision Letter · Decision Letter 1]

7 Oct 2024

Dear Dr. Asada,

Thank you very much for submitting your manuscript "Critical role of Babesia bovis spherical body protein 3 in ridge formation on infected red blood cells" for consideration at PLOS Pathogens. As with all papers reviewed by the journal, your manuscript was reviewed by members of the editorial board and by several independent reviewers. The reviewers appreciated the attention to an important topic. Based on the reviews, we are likely to accept this manuscript for publication, providing that you modify the manuscript according to the review recommendations.

One reviewer has identified a few minor changes/suggestions to improve clarity. I do not anticipate that these modifications will require significant time or effort to correct and I do not plan to send the paper out for another round of review.

Sincerely,

Kirk W. Deitsch

Academic Editor

PLOS Pathogens

James Collins III

Section Editor

PLOS Pathogens

Michael Malim

Editor-in-Chief

PLOS Pathogens

orcid.org/0000-0002-7699-2064

Reviewer Comments (if any, and for reference):

Reviewer's Responses to Questions

**Part I - Summary**

Reviewer #2: N/A

**Part II – Major Issues: Key Experiments Required for Acceptance**

Reviewer #2: 1. The authors have satisfactorily addressed the reviews comments. Additional analyses have been performed with statistical support to back up the authors conclusions.

**Part III – Minor Issues: Editorial and Data Presentation Modifications**

Reviewer #2: 2. Line 162. It was not clear to me how the gold particles were counted. Does n=20 mean that the number of gold particles were counted per ridge for 20 ridges and per spherical bodies for 20 of these, etc? If so, please make this clearer.

3. Line 187. For 1D Tg1-, the bar goes up to 0.7. Is this because its density is 0.7 compared to the MSA2a2 band which is one? Please be clearer on how the densities are normalised.

4. Line 250, what is anti-BSP4?

5. Line 277. Consider indicating where SB are using arrowheads.

6. Why are BBEC called BBMEC on graph 5B?

7. Line 317. What about VESA1 expression?

8. Line 362, Should S14 be S15? There could be other figure labelling mistakes so check them all.

9. Line 460. Is “exportation” a real word? Just say “export”

PLOS authors have the option to publish the peer review history of their article (what does this mean?). If published, this will include your full peer review and any attached files.

Reviewer #2: No

Figure Files:

Data Requirements:

Reproducibility:

References:

---

## [Editor Report · Decision Letter 2]

18 Oct 2024

Dear Dr. Asada,

We are pleased to inform you that your manuscript 'Critical role of Babesia bovis spherical body protein 3 in ridge formation

on infected red blood cells' has been provisionally accepted for publication in PLOS Pathogens.

Best regards,

Kirk W. Deitsch

Academic Editor

PLOS Pathogens

James Collins III

Section Editor

PLOS Pathogens

Michael Malim

Editor-in-Chief

PLOS Pathogens

orcid.org/0000-0002-7699-2064
---

## [Editor Report · Acceptance letter]

25 Oct 2024

Dear Dr. Asada,

We are delighted to inform you that your manuscript, "Critical role of Babesia bovis spherical body protein 3 in ridge formation
on infected red blood cells," has been formally accepted for publication in PLOS Pathogens.

Best regards,

Michael Malim

Editor-in-Chief

PLOS Pathogens

orcid.org/0000-0002-7699-2064